# Immune Response to SARS-CoV-2 Vaccines

**DOI:** 10.3390/biomedicines10071464

**Published:** 2022-06-21

**Authors:** Navya Bellamkonda, Upendra Pradeep Lambe, Sonali Sawant, Shyam Sundar Nandi, Chiranjib Chakraborty, Deepak Shukla

**Affiliations:** 1Department of Ophthalmology and Visual Sciences, University of Illinois at Chicago, Chicago, IL 60612, USA; navyab3@uic.edu; 2ICMR-NIV, Mumbai Unit, A. D. Road, Parel, Mumbai 400012, India; upendralambe69@gmail.com (U.P.L.); ercintsonali@gmail.com (S.S.); 3School of Life Science and Biotechnology, Adamas University, Kolkata 700126, India; drchiranjib@yahoo.com; 4Department of Microbiology and Immunology, University of Illinois at Chicago, Chicago, IL 60612, USA

**Keywords:** COVID-19, SARS-CoV-2, mRNA vaccine, adenoviral vectored vaccine, inactivated vaccine, subunit vaccine, adjuvants

## Abstract

COVID-19 vaccines have been developed to confer immunity against the SARS-CoV-2 infection. Prior to the pandemic of COVID-19 which started in March 2020, there was a well-established understanding about the structure and pathogenesis of previously known Coronaviruses from the SARS and MERS outbreaks. In addition to this, vaccines for various Coronaviruses were available for veterinary use. This knowledge supported the creation of various vaccine platforms for SARS-CoV-2. Before COVID-19 there are no reports of a vaccine being developed in under a year and no vaccine for preventing coronavirus infection in humans had ever been developed. Approximately nine different technologies are being researched and developed at various levels in order to design an effective COVID-19 vaccine. As the spike protein of SARS-CoV-2 is responsible for generating substantial adaptive immune response, mostly all the vaccine candidates have been targeting the whole spike protein or epitopes of spike protein as a vaccine candidate. In this review, we have compiled the immune response to SARS-CoV-2 infection and followed by the mechanism of action of various vaccine platforms such as mRNA vaccines, Adenoviral vectored vaccine, inactivated virus vaccines and subunit vaccines in the market. In the end we have also summarized the various adjuvants used in the COVID-19 vaccine formulation.

## 1. Introduction

COVID-19 vaccines are designed to confer acquired immunity against the severe acute respiratory syndrome coronavirus 2 (SARS-CoV-2). Previous outbreaks of similar coronaviruses, such as SARS and the Middle East Respiratory Syndrome (MERS) have provided valuable information to guide research on the structure and pathogenesis of the current SARS-CoV-2. Early in the year 2020, this understanding aided the creation of various vaccination platforms [1]. SARS-CoV-2 vaccinations were developed with the goal of preventing symptomatic and severe sickness [2]. The first successful sequencing of the genetic data for the betacoronavirus originating from Wuhan occurred on 10 January 2020 by GISAID and was followed by over 320,000 additional sequences from samples of coronaviruses from around the world [3]. On 19 March 2020, the global pharmaceutical sector affirmed their commitment to addressing the COVID-19 situation.

The various COVID-19 vaccines have demonstrated great efficacy (Table 1) in reducing the severity and fatality of the disease [4]. The COVID-19 vaccines have been extensively credited with decreasing the severity and fatality rates associated with COVID-19. Various nations’ programs for vaccine distribution have been widespread and focused on prioritizing at-risk individuals and frontline workers such as healthcare personnel. According to official estimates from national public health organizations, 11.8 billion doses of COVID-19 vaccinations had been administered worldwide as of 17 March 2022, though only a small percentage of these have been distributed in low-income countries [5].

Generally, vaccination efforts for diseases predating COVID-19 spanned years and even decades and no vaccine had been developed to address infectious diseases in such a short time frame [6]. Prior to the COVID-19 vaccine, vaccines that targeted coronaviruses in animals, such as Bovine CoV, and infectious bronchitis virus in chickens, were developed and approved for use by the FDA [7]. Some research was conducted on vaccine development for other viruses in the Coronaviridae family that affect humans such as Severe Acute Respiratory Syndrome (SARS) and Middle East Respiratory Syndrome (MERS), though interest in this research waned as the prevalence of these viruses dissipated [8]. Non-human animals have been used to test vaccines against SARS [9] and MERS [10].

**Table 1 biomedicines-10-01464-t001:** Reported efficacy of SAR-Cov-2 vaccine candidates.

Sr. no.	Vaccine	Efficacy	Side Effects
1	Pfizer BioNTech	N = 43,000 Overall efficacy 95% Preventing severe disease 87.5% [11]	Rare allergies and anaphylaxis
2	Moderna	N = 30,000 Overall efficacy 94.5% Preventing severe disease 100% [12]	Rare facial paralysis (Bell’s Palsy)
3	Astra—Zeneca Oxford	Overall efficacy 70% (64% after 1 dose) (70.4% after 2 doses) [13]	Rare thromboembolic events, rare cases of blood-clots, pulmonary embolism, thrombocytopenia
4	Janssen Johnson & Johnson	72% in the USA 66% in Latin America 57% in South Africa [14]	Rare cases of blood clots, thrombocytopenia, Guillain-Barré Syndrome
5	Sputnik V	91.6% [15]	No serious side effect
6	CoronaVac	COVID 19 Prevention: 65% Severe disease: 86–90% [16]	No serious side effect
7	Covaxin	COVID-19 prevention: 78% Severe disease: 93% Adults aged < 60 years: 79% Old Aged > 60 years: 68% [17]	Headache, fever, fatigue, muscle aches, nausea, pain, irritation, redness, and swelling at the site of the injection
8	Sinopharm BBIBP-CorV	COVID-19 prevention: 65% Severe disease: 86–90% [18,19]	No serious side effect
9	Epivac	Phase III clinical trials: 79% [20]	Data not available
10	MVC COVID-19 vaccine	Overall efficacy: 84% [21]	No vaccine-related Serious Adverse Reaction
11	Zifivax	Efficacy against COVID-19 Short-term: 81.4% Long term: 75.7% [22,23]	Injection site pain, headache, fatigue, fever, body ache, abdominal pain, nausea and vomiting,
12	CorbeVax	Efficacy: Phase II & III clinical data is pending [24]	Fever/Pyrexia, Headache, Fatigue, Arthralgia, Urticaria, Chills, Injection site swelling
13	Soberana 02	First shot: 71% Second shot: 92.4% [25]	No serious side effect, fewer than 1% of participants in the phase III trial developed a fever

Research in 2005 and 2006 was conducted in order to find a solution to the growing prevalence of SARS [26,27,28,29]. There has yet to be developed a safe and effective treatment or protection strategy against SARS [30,31]. Upon the occurrence of the major MERS outbreak in 2012, previous research on SAR-CoV allowed for more efficient research into the pathogenesis and structure of MERS-CoV due to their similarities [32,33], though no effective vaccine was developed during that period or ever since [34].

Viral-vectored candidates for MERS vaccines in humans did reach Phase I clinical trials with two being two adenoviral-vectored, and one MVA-vectored [35,36]. Vaccines made with an inactive or weakened virus often take over a decade to produce. mRNA was considered a candidate for more rapid vaccine development as mRNA molecules can be generated more quickly. As early as 2012, scientists were able to employ mRNA therapy to induce erythropoiesis in mice as a treatment for certain diseases [37]. Human trials for a human mRNA treatment developed by Moderna targeting oncology and infectious diseases began in 2015 [38]. After the technique was cleared for Ebola, viral vector vaccines were created for the COVID-19 pandemic [39].

Universities, governments, public health organizations, and biotech firms all worked rapidly to develop different COVID-19 vaccines using various licensed techniques which were evaluated using observational and case-control studies in order to measure real-world vaccination efficacy (RWE) [40]. These studies focused on both the short-term and long-term effectiveness of mRNA vaccines targeting SARS-CoV-2 are being investigated in a study [41].

## 2. Vaccine Development Approaches

Early into vaccination development efforts, the coronavirus spike protein (S protein) was identified as a promising target due to its ability to trigger significant B-cell and T-cell immune responses [42,43,44]. Other coronavirus proteins, such as the nucleocapsid (N) proteins, are also being studied for vaccine development since they are able to generate a strong T-cell response and their genes are frequently conserved and less prone to recombination compared to spike proteins [45,46].

Various platforms developed in 2020 include non-replicating viral vectors, nucleic acid technologies, live attenuated viruses, and inactivated viruses [42,47,48]. Viral vector vaccine technologies such as those used in the development of the influenza vaccine were not popularly employed in COVID-19 vaccine technologies as newer techniques were proven to better utilize the pathogenesis of COVID. In order to prevent the conversion of prefusion spike proteins into their elongated form, a 2P mutation is used to stabilize the protein’s form [49]. This protein would still trigger an adaptive immune response in human cells to the virus before the protein could bind to the cell [49].

## 3. Immune Response Generated by SARS-CoV-2 Infection

Coronaviruses are characterized by their large, protruding spike proteins that form a 100-nanometer crown, or corona, around the virus [50]. The envelope is comprised of four structural proteins, envelope (E), spike (S), membrane (M), and nucleoprotein (N), a lipid bilayer obtained from the host’s cell membrane, as well as a variable number of nonstructural proteins. SARS-CoV-2 recognizes and the class I fusion spike proteins attach to angiotensin-converting enzyme 2 (ACE2) commonly found in the respiratory epithelial cells of the host [51,52]. This allows for the viral capsid to fuse with the host cells and inject its nucleic acid into the host cells. The S2 subunit plays a vital role by permitting viral and cellular membranes to fuse and the S1–S2 junction is cleaved by the serine transmembrane protease serine 2 (TMPRSS2) found in the host [52]. In cats and ferrets, SARS-CoV-2 replicates well, but not in dogs, pigs, chickens, or ducks with cats having recently been shown to be especially susceptible to an experimental airborne illness [53]. Recent research has shown that the same receptor may be involved in infection in both cats and ferrets and is mediated by the transmembrane spike (S) [54]. Following attachment to the cell surface, the virus may enter the cells through endocytosis [55]. In vitro and in vivo research has been conducted regarding the translation and budding processes of SARS-CoV-1 and MERS viruses [56]. Different intercellular sensors, such as RIG I/MDA5/MAVS/TRAF3/IRF3/IRF7 and various TLRs/TRIF/MyD88/IkB/NF-kB/MAPK/AP-1 pathways, detect SARS-CoV-1 infection [57]. RIG I and IRF3/7 pathways are involved in type 1 interferon responses and may be inhibited by SARS proteins resulting in reduced antiviral response while boosting NF-kB activation, pro-inflammatory cytokine production, and necroptosis [57]. These signaling events have been implicated in hyper inflammation, increased cellular death, and cytokine storms [50].

The innate immune system possesses immunological memory, dubbed “trained memory”, which can influence the severity of the illnesses. The complex pathogenicity of COVID-19 and the disease’s virulence are linked to viral activation of the cytoplasmic NOD-like receptor family, pyrin domain containing 3 (NLRP3) inflammasome [58]. Macrophage and epithelial cell activation leads to the production of pro-inflammatory cytokines such as interleukin (IL)-1 and IL-18, which contribute to the pathogenic inflammation that causes COVID-19 symptoms to be so severe [59].

Toll-like receptors TLR3, TLR7, TLR8, and TLR9 detect mRNA, which leads to the induction of the NF-kB inflammatory pathway and a large number of pro-inflammatory cytokines that play a key role in the initiation of virus-induced inflammation [60,61]. Aside from heightened levels of acute-phase reactants and cytokine storm, little is known about the innate immune response. Presently, the majority of reports have concentrated on serious consequences and adaptive immune responses.

SARS-CoV-2 viral proteins attack a number of innate immune signaling proteins. Nsp13, Nsp15, and open reading frame ORF9b target the interferon (IFN) pathway, while Nsp13 and Orf9c target the NF-B pathway. SARS-CoV-2 Orf6 hampers NUP98-RAE1, an interferon-inducible mRNA nuclear export complex [62]. SARS-CoV-2 replication proteins Orf3b and Orf9c are canonical.

### B and T Cell Immune Response

More than 80% of individuals with SARS-CoV-2 infection have a conventional respiratory virus-like clinical course that is mild to severe and self-limiting [50]. SARS-CoV-2 spike proteins bind to the cluster of differentiation 147 (CD147) and infect human T-cell lines and this pathway may even be implicated in inducing T-cell death [63]. CD147 is found in a variety of tissues and cells and is involved in cell proliferation, apoptosis, tumor cell migration, metastasis, and differentiation, particularly in hypoxic environments [64]. CD147 Ig0-Ig1–Ig2 is a second isoform of CD147 that has been identified. CD147 dysregulation has been linked to many varieties of cancer and regulates the production of matrix metalloproteinase (MMP) and vascular endothelial growth factor (VEGF), as well as signals for tumor cell invasion and metastasis [65]. COVID-19 treatment may be prevented by hindering the SARS-CoV-2 spike binding and subsequent infection by blocking CD147 protein with meplazumab [64].

In order to proliferate, viruses attach to cells that express specific receptors and then are able to enter the cells. In order to combat these viruses, class I major histocompatibility complex (MHC) proteins present viral peptides to CD8+ cytotoxic T. The virus-infected tissue cells are lysed by CD8+ cytotoxic T lymphocytes. Other antigen-presenting cells such as dendritic cells and macrophages present viral particles via MHC-Class-II to CD4+ T lymphocytes [66]. These CD4+ T cells, or helper T-cells, can interact with B cells which can identify viruses directly and produce virus-specific antibodies. Within the first week, after the onset of symptoms, the IgM isotype primary virus-specific antibody response is detected [67]. This can generate lifelong immunity through the production of IgG isotype antibodies that follow the early IgM response.

T- and B-cell responses to viral determinants, as well as identification of virus-infected cells, are crucial to our knowledge of various antiviral immune events. The S1 and S2 subunits of the Coronavirus spike protein are broken at the S1/S2 border and the S2′ cleavage site. SARS-CoV-2 spike proteins polybasic furin cleavage sequence (PRRARS) differs by four-amino-acid insertion from those in other SARS. T- and B-cell epitopes have been identified that demonstrate immunodominance [68]. SARS-CoV-2 is the third SARS-like virus to cause an outbreak, and thus shares many similarities with SARS-CoV-1 and MERS-CoV. These similarities may have resulted in those populations affected by the 2002 and 2012 outbreaks having some immune protection from COVID-19. Potential T-cell and B-cell epitopes of SARS-CoV-2 have been predicted using a bioinformatics technique based on their resemblance to SARS-CoV possible T-cell and B-cell epitopes, though only SARS-CoV T-cell epitopes and 16% of known SARS-CoV B-cell epitopes map to SARS-CoV-2 [69,70]. The conserved B- and T-cell epitopes shared between SARS-CoV and SARS-CoV-2 show the potential to lead to vaccination strategies that direct the immune response to these conserved regions. This immunity not only builds on existing research, but may result in cross-protective immunity against other varieties of coronavirus and future strains that may be mutated [69]. SARS-CoV-2 structural proteins show significant genetic similarities with SARS-CoV but not to MERS-CoV. Reactivity of SARS-specific memory T-cells was found to persist for 9- to 11-years post-infection and was specific to the S, N, and M proteins in SARS-type viruses with no cross-reactivity with MERS [71].

## 4. mRNA Vaccines

mRNA vaccines have been considered for over three decades and are comprised of self-replicating RNA or messenger RNA that cause cells to express an antigen that can elicit cytotoxic T lymphocyte responses. mRNA shows specific benefits in inducing transient expression and production of specific antigenic proteins that give rise to MHC-1 presentation [72]. This has made RNA a popular choice in several COVID-19 vaccines, including the approved Pfizer–BioNTech and Moderna vaccines. These vaccines use RNA to induce the expression of SARS-CoV-2 spike protein which are then recognized and destroyed, allowing the body to build up memory immune cells. Nucleoside-modified messenger RNA is commonly used in RNA vaccines, although this is not necessarily always the case. The mRNA molecules are combined with lipid nanoparticles which improve the absorption of RNA into cells and protect the strand while in transport [73,74,75]. The first COVID-19 vaccines to be approved in the United Kingdom, the United States, and the European Union were RNA vaccines [76,77].

Severe allergic responses are uncommon. In December 2020, of the 1,893,360 people who received the first dose of the Pfizer–BioNTech COVID-19 vaccine, there were 21 cases of anaphylaxis [78]. In total, 10 occurrences of anaphylaxis were reported between December 2020 and January 2021 of the 4,041,396 Moderna COVID-19 vaccine dosages administered. The allergic symptoms were most likely caused by lipid nanoparticles (LNPs) [79].

### 4.1. Mechanism of Action

A vaccine’s purpose is to drive the body to produce pathogen-specific antibodies as its adaptive immune response. These antibodies target the antigens on or created by the pathogen [80]. This response may be elicited through the use of attenuated, or inactivated forms of the virus or recombinant antigen-encoding viral vector which are harmless carrier viruses with an antigen transgene. Conversely, mRNA vaccines use a synthetic, transient segment RNA encoding a viral antigen [81]. Dendritic cells pick up these mRNA snippets through phagocytosis [82]. Dendritic cells read the mRNA and make the viral antigens that the mRNA encodes using their own machinery (ribosomes). Within a few days of their introduction, the mRNA fragments are degraded by the body [68]. Dendritic cells take vaccine mRNA globules far more quickly than non-immune cells, which can also absorb vaccine mRNA, generate antigens, and show antigens on their surfaces. Fragments of mRNA are translated in the cytoplasm and have no effect on the genomic DNA stored in the cell nucleus [83].

After the host cell produces viral antigens, the adaptive immune system goes through its typical procedures. Proteasomes are enzymes that break down antigens. The antigen eventually attaches to MHC molecules which are able to move to the cellular membrane and activate dendritic cells when bound to an antigen (Figure 1). Dendritic cells travel to lymph nodes after activation and present the antigen to T- and B-cells, resulting in the production of antibodies that target the antigen [84].

### 4.2. B Cell and Antibody Response

Most licensed vaccines protect against viral infection through the induction of long-lasting protective antibody (Ab) responses from activated B-cells [85]. The activation of B-cells by an antigen presented by an MHC molecule leads to the production of plasmablasts (PB) in extrafollicular (EF) sites as a part of the short-term immune response. Following this, germinal centers (GCs), microanatomical locations of secondary lymphoid organs, produce affinity-matured, persistent antibodies against viruses [86]. The GC is the site where activated B-cells undergo random mutations. This produces Abs with a higher affinity for the antigen. B-cells with stronger affinities are selected for, resulting in higher concentrations of more specific B-cells. This process creates high-affinity long-lived plasma cells (LLPCs) and memory B-cells (MBCs). LLPCs release antibodies, some of which are neutralizing antibodies (nAbs), which may play a role in sterilizing immunity [87]. These LLPCs are important for their varying ability to survive for many years in some situations, secreting Abs even with no additional antigen activation [88]. In the case of a future pathogen encounter, MBCs become activated and increase the production of high-affinity Ab-secreting cells. Thus, it is desirable for vaccinations to induce the production of LLPCs and MBCs.

Individuals infected with SARS-CoV-2 have been shown to develop antibodies that target SARS-CoV-2 S proteins in a process that may reduce SARS-CoV-2 infection in vitro and/or in vivo, according to studies of natural infection in humans [89,90]. SARS-CoV-2 mRNA vaccine efficacy has been determined in pre-clinical assessments by focusing on their abilities to elicit binding and neutralize Ab responses in mice. Importantly, researchers discovered that the Ab responses elicited by these mRNA-1273 (Moderna), BNT162b2 (BioNTech/Pfizer), and CVnCoV (CureVac (Frankfurt, Germany)) SARS-CoV-2 mRNA vaccines were able to neutralize the virus in vitro, as evaluated by both pseudovirus and SARS-CoV-2-based neutralization assays, Refs. [91,92,93,94,95]. Appropriate nAb levels were observed greater than two months following administration of these mRNA vaccines. Further analysis of the Ab responses to the mRNA vaccines showed that the Ab response was boosted following the administration of a booster injection after initial priming [93,95,96]. Higher vaccine doses (30 g) appeared to be adequate with only a single dose to generate an increased humoral response in mice, however, at lower vaccine doses of 1 g or 2 g, a second round of immunization was required for sufficient nAb production. Overall, the findings from these many mice investigations imply that in order to reach nAb levels following immunization with SARS-CoV-2 mRNA vaccines, two doses or a single higher dose may be required. The choice of antigen has been shown to affect Abs and nAbs production as a study found that receptor-binding domain (RBD) mRNA vaccines were more efficient at generating SARS-CoV-2-binding antibodies compared to SARS-CoV-2 S1 mRNA vaccines [97]. Significantly stronger humoral responses were found in studies with mRNA vaccines encoding the full-length S protein [97].

The SARS-CoV-2 has some species specificity, which limits studies due to the wild-type virus being unable to replicate in laboratory mice. Thus, research involving non-human primates (NHPs) is important for determining the efficacy of vaccine candidates [98]. Two clinical candidates for a SARS-CoV-2 vaccine, mRNA-1273 (10 or 100 g) and BNT162b2 (30 or 100 g), both showed the dose-dependent ability to induce the production of antibodies targeting SARS-CoV-2 proteins. In vitro, the values evoked in NHPs by BNT162b2 were higher than the nAb values generated from SARS-CoV-2 convalescent human sera [93,94]. Larger doses of mRNA vaccines were required to evoke sufficient amounts of S- and RBD-binding antibodies when only a single immunization was used [95]. Utilization of a booster immunization showed that these vaccine candidates resulted in in vivo protection against SARS-CoV-2 in all cases. Although these vaccine candidates did not prevent viral replication in the upper respiratory tract in animals, the virus was only detected in these areas for a few days following infection. Fluid and nasal swabs of the bronchoalveolar lavage fluid in primates following infection after immunization showed the absence of significant SARS-CoV-2 subgenomic RNA which provided evidence to indicate that the lower respiratory tract of primates following immunization was protected from viral infection [93]. The CVnCoV candidate was evaluated with several lower doses (0.5 or 8 g), and studies showed that only the higher dose of the ones tested evoked significant levels of RBD and S-binding IgG titers following a single immunization, with the amount of IgG increasing in the time post-immunization [80]. After the booster immunization, these candidates’ nAb responses were shown to be accompanied by in vivo protection against SARS-CoV-2. These trials also showed that higher vaccine doses provided better upper respiratory tract protection, but both doses provided equal protection for the lower respiratory tract [93,94].

When it comes to the development of coronavirus vaccines, there are certain noticeable concerns about the quality of antibody manufacturing. One concern is that currently proposed SARS-CoV-2 vaccines may result in antibodies with low specificity for the virus or Abs that are unable to destroy the virus [6]. No evidence exists currently to prove the significant presence of this Ab-dependent enhancement occurring following SARS-CoV-2 infection in humans.

### 4.3. T Cell Response

There are mixed success rates for the effectiveness of the mRNA vaccine candidates in their activation of CD8+ T-cell responses. A study in 2020 reported that a single dosage of SARS-CoV-2 mRNA given to mice elicited CD8+ T-cell production targeting specific SARS-CoV-2 antigens in the spleen and lungs [91]. Furthermore, CD+8 T-cells activated by SARS-CoV-2 mRNA vaccines displayed cytotoxic T cell characteristics (Granzyme B and CD107a). Another 2020 study discovered a significant increase in SARS-CoV-2-specific CD8+ T-cells after immunization with SARS-CoV-2 mRNA, as evaluated by IFN production on CD8+ T-cells following another activation by antigens, in an independent investigation [99]. Recent findings show that CD8+ T-cells may comprise the main immune response shortly after immunization and can be observed in patients as early as one week after vaccination [100].

There were also mixed results regarding the production of cytotoxic T-cells after immunization in animal models examined in preclinical studies. BNT162b2 to mice resulted in an increase in the presence of CD8+ T-cell indicators, IFN- and IL-2, in the spleen 12 days after immunization, a finding that was consistent in rhesus macaques [94]. A 2020 study observed a similar observation in CD8+ splenocytes infected with CVnCoV [95]. Another study from the same year was also able to detect a large growth of CD8+ T effector memory cells in mice immunized with clinical candidate ARCoV [96]. However, in another 2020 study using macaques, the Moderna mRNA-1273 vaccine did not elicit an increase in CD8+ T-cell response as it did in mice [93]. Similar situations occur in humans as the S-protein does not induce a strong CD8+ T-cell response in all naturally infected humans [44]. While the increase of cytotoxic CD8 T-cells may be advantageous, there is no evidence that it is essential for successful SARS-CoV-2 vaccine protection.

### 4.4. Pfizer–BioNTech and Moderna

Lipid nanoparticles (LNPs) are coupled with a prepared mRNA vaccine in the Pfizer–BioNTech (Comirnaty) BNT 162b2 and Moderna (1273) vaccines. Despite the fact that mRNA-based vaccinations have never been licensed before, this work was moved forward at an unparalleled rate [101]. The use of genetically edited RNA to elicit a managed immune response through the production of viral proteins is what makes RNA vaccines so innovative [51]. The synthetic nature of mRNA-based vaccines eliminates the need for cell culture or virus fermentation and thus allows for more efficient development.

Of the 25–28 proteins that SARS-CoV-2 possesses, researchers have only had to isolate the mRNA encoding the spike (S) proteins [102,103]. The mRNA’s sensitivity to degradation makes it necessary to address the limitations of mRNA vaccines. The Pfizer–BioNTech COVID-19 vaccine circumvents this issue by requiring storage at −94° Fahrenheit, or −70° Celsius. This prevents mRNA degradation but requires the use of costly freezer devices which may not be available in low- and middle-income countries (LMCI). Despite this, no adjuvants or preservatives are utilized in these vaccines as the vaccine is self-stimulating. Furthermore, a lipid particle encloses the mRNA in order to prevent degradation.

The mRNA is extracted from the SARS-CoV-2 virus and incorporated into a lipid nanoparticle. The LNP is injected intramuscularly and is eventually able to attach to host cells and inject the mRNA into the cytoplasm (Table 2). The mRNA reaches the ribosomes, which are then used to synthesize the viral spike proteins in a process referred to as translation [51]. The proteins are then recognized by antigen-presenting cells, such as B-cells, macrophages, and dendritic cells, which possess MHC-2 molecules on their surface or cells that possess MHC-1 molecules. The binding of the protein acting as an antigen to the MHC molecules leads to immune recognition of the S-proteins by cytotoxic T-cells in the case of MHC-I cells and helper T-cells (Th) in the case of MHC-II cells. The CD4+ T-cell receptors on Th cells bind to the antigen-presenting MHC-II which triggers the production of cytokines such as IL-2, IL-4, and IL5 [104]. These interleukins cause our body’s B-cells to mature into plasma cells, which create a significant quantity of Abs targeting viral spike proteins. In the meantime, interleukins (IL) result in memory T-cell proliferation. The TCRs of CD8+ T-cells are activated by the presentation of the spike protein antigens by MHC-1 proteins on cell membranes, resulting in the production of cytotoxic T-cells (Tcx) which directly cause the death of virus-infected cells using harmful molecules such as granzyme and perforin [105]. Tcx cells also secrete immune signals to further amplify the immune response [106].

The vaccines elicit a significant reaction, but we do not yet know how long this protection lasts. The ideas are sometimes diametrically opposed. Studies have shown that this immunity should last somewhere between six and nine months [107,108].

The side effects of the vaccine following administration of both the first and second dose have been considered in order to determine vaccine safety. The two doses are given 21 days apart, and any possible symptom is recorded after 7 days from the last dose [106].

Both vaccines have been reported to bear only minor or non-existent adverse effects. A slight pinched sensation at the injection site, some redness, weariness, headache, muscle, and joint pain, and fever are all possible side effects, especially after the second dose. A few other common side effects, including Guillain-Barre syndrome and anaphylactic reactions, have been recorded in the literature for the vaccination Pfizer, though these are extremely rare [78,109]. Of 1,893,360 individuals who received at least a single dose of the Pfizer–BioNTech COVID-19 vaccine, 21 suffered from allergic reactions, including anaphylaxis in some cases. From 10 to 23 December 2020, 11.1 individuals per 1 million doses experienced an allergic reaction to the Pfizer vaccine. The Moderna COVID-19 vaccine has an anaphylaxis rate of one per 2.5 million administered doses. However, no deaths related to anaphylaxis caused by the mRNA COVID-19 vaccines have been reported. A small number of individuals experienced facial paralysis (Bell’s Palsy) following vaccination, however, Bell’s Palsy was not found to be higher after vaccination [78,109].

#### 4.4.1. Performance Evaluation of Pfizer–BioNTech Vaccine

Participants (n = 43,000) were divided into two groups: placebo (saline solution injected twice) and vaccine (two vaccine doses injected). In the placebo group, 162 people reported symptoms, of those, 9 were considered severe. Furthermore, 8 individuals in the vaccine group had symptoms, although only one had severe symptoms. The effectiveness of the vaccine in preventing disease was 95 percent, while the effectiveness against severe disease was 87.5 percent. Both doses were administered 21 days apart, with any side effects appearing seven days following the final dose [110].

#### 4.4.2. Performance Evaluation of Moderna Vaccine

Participants (n = 30.000) were randomized into two groups: placebo (two saline injections) and vaccination (two vaccine doses). There were 185 people in the placebo group (30 showed severe symptoms). In the vaccinated group, 11 tested positive for COVID-19, with no severe symptoms in the group. The efficacy against disease was 94.5 percent in this example, with the efficacy against severe symptoms being 100%. The two doses were given 28 days apart (now, a 42-day interval is advised), and side effects were recorded 7 days following the final dose. The Pfizer vaccine must be stored at −94° Fahrenheit (−70 °C), while the Moderna vaccine must be kept at −4° Fahrenheit (−20 °C) [111,112] (FDAa and b 2021).

## 5. Viral Vector Vaccines

Several COVID-19 vaccines use adenovirus vector carrying DNA that encodes an antigenic SARS-CoV-2 protein. This DNA is injected into the host’s cells and directs the cell to encode for an antigen that stimulates a systemic immune response.

### 5.1. Astra-Zeneca Oxford

The AstraZeneca adenovirus viral vector vaccine was developed in a collaboration between Oxford University and AstraZeneca in the United Kingdom and makes use of a genetically engineered vector carrying genetic information encoding a wild-type S protein. Though first used in the United Kingdom, the vaccine has been made available in over 170 nations. The immune response in the AstraZeneca vaccine is generated from the use of a modified replication-deficient chimpanzee DNA adenovirus, ChAdOx1, that human populations have not been exposed to (Table 2). This assures that most individuals have no prior immunity to the virus from which the vector was taken and the immune response is only reactive towards the viral protein encoded by the vaccine’s injected viral DNA, rather than the adenovirus itself. The DNA encodes a protein that elicits a similar immune response as the SARS-CoV-2 S-peptide along with a tissue plasminogen activator (tPA). In human host cells, this vector leads to the production of new adenovirus viral proteins that generate this immunological response. The adenovirus latches on to the outside of the human host cell and DNA is then injected into the cytoplasm where it travels to the nucleus. The DNA is transcribed into mRNA by the host cells and this RNA is later translated into the appropriate proteins by ribosomes. MHC1 and MHC2 complexes are formed when the proteins are expressed on cell membranes. The processes of RNA and DNA vaccines are identical at this point, leading to the activation of T-cells, B-cells, and plasma cells along with the production of antibodies [113].

The vaccine was found to be safe in early investigations, with very minor side effects such as pain at the injection site, myalgias, headache, redness, and arthralgias. The rate of facial paralysis (Bell’s Palsy) following the administration of the AstraZeneca vaccine was 13 per million doses; however, the vaccine was not found to be the cause of these side effects. The majority of the 143 people who died soon after immunization were geriatric individuals with pre-existing health conditions, therefore these fatalities were likely not linked to the vaccine. The vaccine’s side effects, on the other hand, are not yet fully understood. Rare thromboembolic events, such as cerebral venous sinus thrombosis [CVST], were reported following vaccination which raised concerns about vaccines [114].

These side effects forced the European Medicines Agency (EMA) to remove the vaccine from circulation on 15 March 2021, but it was approved again three days later, with the EMA declaring that the vaccination’s benefits outweighed its dangers. Furthermore, the EMA emphasized that there is no indication of concerns with individual vaccine batches. Furthermore, of the 20 million individuals in Europe who received the vaccine, 7 developed disseminated intravascular coagulation, and according to the organization (DIC) 18 presented CVST (with 9 deaths) [114,115]. The majority of these symptoms were seen in women under the age of 55. Due to these concerns, AstraZeneca has been banned in Germany and several European countries for adults under the age of 60. Although a causal link between these disorders and the vaccine has yet to be proved, it has been determined that the occurrence warrants further investigation.

In the case of this vaccine, the capacity to prevent infection in its initial stages and not the disease, was used to determine the efficacy of this vaccination. This entailed determining whether SARS-CoV-2 was still replicating in the body following vaccination (infection), as opposed to, a reduction or persistence of symptoms (disease). Even those who are infected without the presentation of symptoms are able to shed the virus and thus transmit it to others. Asymptomatic individuals were found and the virus’s transmissibility was assessed. On the investigated samples, RT-PCR was used to detect infection transmission. The two doses were administered 28 days apart (12-week intervals are under study). PCR was used to detect the existence of infections after 14 days. Exploratory analysis revealed that longer prime-boost intervals result in increased vaccine efficacy, as well as the fact that a single vaccine dosage is effective for the first 90 days, bolstering current policy [116]. AIFA approved the vaccine’s use for persons 65 and older, due to health concerns regarding blood clotting overserved in younger recipients, on 18 February 2021, primarily in countries with circulating virus strains [117].

A Brazilian study found that the vaccine’s efficacy was around 62 percent, whereas another UK study found the efficacy to be around 90 percent after two doses. The combined data from both nations contain information on around 12,000 participants from both investigations. The combined efficacy value for both studies was 70% with 0 cases with severe symptoms [118]. The efficacy of the vaccine was reported by the WHO to be 76% 15 days after the second dose is administered. The vaccine has been determined to be acceptable for low- and middle-income nations due to its minimal storage requirements and over 2.5 billion doses have been distributed globally.

### 5.2. Johnson and Johnson (Janssen)

Janssen Pharmaceuticals (Johnson & Johnson, USA) has developed an experimental COVID-19 vaccine that appears to show some success in the prevention of moderate and severe disease in adults. JNJ-78436725, or Ad.26.COV2.S, is a single-dose vaccination (Table 2). It does not require the extreme cold storage of other vaccines, making it suitable for areas without advanced refrigeration. This vaccination works by using a laboratory-made adenovirus vector to deliver a gene encoding for the SARS-CoV-2 spike protein. On 27 February 2021, the Food and Drug Administration in the United States granted an emergency use authorization (EUA) for this vaccine to be used in people aged 18 and over. On 12 March 2021, Italy approved the vaccine based on its efficacy in the prevention of severe symptoms and reducing hospitalizations and deaths. The vaccine demonstrated 77% efficacy against disease after 14 days and 85% after 28 days of treatment [119]. Data currently available suggest that people beyond the age of 65 have equal efficacy. In April, the Italian government approved this vaccination. It was used for a while before being withdrawn due to serious adverse effects, but was later reintroduced to the market.

This vaccine combines the genes for the SARS-CoV-2 spike proteins with modified DNA of the respiratory virus adenovirus 26 that has been shown to not harm humans (adenovirus 26 CoV2). Similar to the AstraZeneca vaccine, a modified adenovirus vector with genetic information encoding the S-protein in viral DNA is injected intramuscularly [120]. The DNA enters the cytoplasm and then travels to the nucleus and is transcribed into mRNA which can be translated to have the host generate spike proteins. The DNA remains distinct from the host cell’s DNA. The viral spike proteins present themselves on the surface of the host cells which leads to immune recognition and triggers the formation of CD4+ and CD8+ T-cells, plasma cells, interleukin (IL), and B-cells [44]. T-cells are suited to targeting infected host cells whereas antibodies are effective at targeting free viral particles, preventing further infection.

During the delivery of this vaccination, minor adverse effects such as swelling, redness, and discomfort were reported at the injection site. A day after the CDC granted the EUA allowing the administration of the Johnson & Johnson COVID-19 vaccine, the CDC and FDA suggested a “pause” in the administration of the vaccine, following the report of six incidences of severe varieties of blood clotting in people who received the vaccine in the United States. These instances of cerebral venous sinus thrombosis (CVST) with thrombocytopenia, were similar to those cases reported following administration of the AstraZeneca vaccine [121]. This resulted in the vaccine’s usage being halted in Italy, despite the fact that it had only recently been certified for emergency use.

A significant benefit of the Janssen Pharmaceuticals vaccine is that it only requires a single shot, meaning it requires half the supply and storage space of two-dose vaccines. These benefits could make vaccine distribution easier and increase vaccine access both domestically and internationally. Currently, the vaccine is used in Russia, but many other countries throughout the world are demanding authorization to produce it.

In total, 468 of 44,325 individuals from Argentina, Brazil, Chile, Colombia, Mexico, Peru, South Africa, and the United State tested positive for symptomatic forms of COVID-19. The Johnson & Johnson vaccine was found to be 66% effective at preventing moderate and severe disease 28 days after the vaccination. There were three deaths in the vaccination group but these were found to not be the result of COVID-19.

Janssen Pharmaceuticals has determined that the vaccine is 85 percent effective in preventing severe illness and that so far, it has been 100% effective at preventing death from COVID-19. Within 49 days of receiving a single short, antibodies were observed and no deaths or hospitalizations occurred in these individuals [122,123,124].

### 5.3. Sputnik V

This vaccine was produced by Gamaleya Research Institute in Russia and has been used there in emergency situations. The Russian Health Ministry registered the Sputnik V vaccination against COVID-19 on 11 August 2020, making it the world’s first. Other countries are attempting to produce it. Hungary was the first European country to approve the vaccine for emergency use, with the National Institute of Pharmacy and Nutrition approving it. Serbia, Russia, Argentina, Algeria, Bolivia, Venezuela, Armenia, Paraguay, the United Arab Emirates, Tunisia, Iran, Guinea, Turkmenistan, and the Palestinian territories have all approved the vaccination. The Sputnik V vaccine has been given to almost 2 million people around the world thus far. The EMA and AIFA are looking at the possibility of approving this vaccine for emergency use in Italy.

This vaccine uses two separate adenovirus vectors (Ad 26 and Ad 5) that may produce a better and extended immune response relative to vaccines with a single vector in both doses (Table 2). Furthermore, according to Kirill Dmitriev, CEO of the Russian Direct Investment Fund, the vaccine is half the cost of other vaccinations with the same efficacy rate.

The Sputnik V vaccine functions similarly to the Johnson & Johnson and AstraZeneca vaccines. The adenovirus vector infects the cells with an engineered DNA fragment encoding for SARS-CoV-2 S protein that is capable of eliciting an immune response resulting in the activation of T-cells and the production of antibodies that can shield people from subsequent infections. Adenovirus vector cannot multiply in human cells.

Sputnik V has no significant negative effects, though some have reported mild flu-like symptoms. The two-dose Sputnik V contains two distinct adenoviruses, adenovirus vector 26 in the first dose and adenovirus 5 in the second dose. Even before the pivotal Phase 3 trials, the vaccine was quickly delivered to humans. Of 19,866 participants in phase III trials, 14,964 received two vaccine doses, and 4902, or about a quarter, received the placebo [125]. Symptomatic COVID-19 was found in 16 cases in the vaccine group 21 days following the first dose, compared to 62 in the placebo group, resulting in a 91.6 percent effectiveness rate. A stronger immune response could be the result of using different adenovirus vectors in each of the doses [126].

The study from Italy has reported the efficacy of the vaccine to be greater than 90% for disease and 100% for severe disease, according to findings.

## 6. Inactivated Vaccines

Inactivated vaccines use a dead version of the pathogen and are generally the quickest choice for antiviral immunizations. Inactivated viruses are promising because they exhibit numerous viral proteins for immune identification, have consistent expression of conformation-dependent antigenic epitopes, and can be mass-produced easily. Historically, inactivated viruses have been utilized for the production of vaccines and have been found to be useful in preventing viral infections such as polio [127]. To date, different chemical and physical strategies have been used to inactivate coronaviruses, including the use of formalin, formaldehyde, -propiolactone, and UV alone or a blend of these techniques [128].

Virus-neutralizing antibodies are primarily important for the protection against viral infection, a principle that remains true for the great majority of viral infections against which humans develop significant immune protection as a result of infection or vaccination. There are already many vaccine candidates that are being considered for their ability to induce the production of antibodies able to bind and neutralize the coronavirus spike protein [129]. Currently available inactivated vaccines serve to improve immunity by eliciting the production. Antibodies of this type frequently disrupt the virus from undergoing conformation changes or its interactions with cell entry receptors [130]. The mechanism of action by which the immune response is generated by inactivated vaccine is depicted in Figure 1.

Adjuvants are necessary to induce an effective and strong immune response because these vaccines provide weaker immunity than live vaccines. Adjuvants are frequently required when inactivated vaccinations are delivered, and periodic booster doses are required to ensure long-term protection, which are the main disadvantages of inactivated vaccines.

### 6.1. CoronaVac

CoronaVac, or the Sinovac COVID-19 vaccine, is an inactivated viral COVID-19 vaccine created by the Chinese biotechnology firm Sinovac. A strain of SARS-CoV-2 from China was selected and grown in Vero cells. The virus was deactivated using beta-propiolactone, which attaches to the genes and leaves the virus and its proteins largely intact. The inactivated virus was adjuvanted with aluminum hydroxide (Table 2). Sinovac’s Phase II trials involving 600 patients given either 3 μg or 6 μg doses with a booster received either 2 or 4 weeks after priming revealed the vaccine was safe in healthy adults, with the 3 ug dose eliciting 92.4 percent seroconversion after 14 days and 97.4 percent after 28 days [128].

### 6.2. BBIBP-CorV

The BBIBP-CorV or the Sinopharm BIBP COVID-19 vaccine is another inactivated viral COVID-19 vaccine created by Sinopharm’s Beijing Institute of Biological Products in China. BBIBP-CorV is an inactivated vaccine developed by growing a virulent SARS-CoV-2 strain in Vero cells and then isolating and inactivating the virus using β-propiolactone and adjuvating it with aluminum hydroxide [131] (Table 2). It was proven to be safe and tolerable in all dosages tested and in both age groups. On day 42, all vaccination recipients had humoral immune responses against SARS-CoV-2. Furthermore, this study found that a two-shot immunization with 4 g vaccine on days 0 and 21 or days 0 and 28 resulted in stronger neutralizing antibody titers than a single 8 g dose or a 4 g dosage on days 0 and 14 [127].

### 6.3. WIBP-CorV (Sinopharm)

WIBP-CorV SARS-CoV-2 is another vaccine produced by Sinopharm. It was produced similarly to other COVID-19 inactivated vaccines which were produced in Vero cells and inactivated using β-propiolactone before being adsorbed onto aluminum hydroxide (Table 2). Phase I and II clinical trials showed that by the 14th day after the second dose, 100% of individuals had binding antibodies against entire inactivated SARS-CoV-2 and 98 percent had neutralizing antibodies, with only minor side effects [127].

### 6.4. Covaxin

The Covaxin vaccine developed by Bharat Biotech in collaboration with the Indian Council of Medical Research is a two-dose inactivated virus-based COVID-19 vaccine. In a manner similar to the Sinopharm and Sinovac vaccines, a sample of SARS-CoV-2 is grown on large amounts of Vero cells. The viruses are then purified and immersed in beta-propiolactone to deactivate them while allowing the virus to remain whole. The inactivated viruses are subsequently adjuvanted with Alhydroxiquim-II (Table 2). Bharat Biotechnology claims an overall 65% efficacy against the DELTA variant with a 98% efficacy in the prevention of severe COVID-19 symptoms [132]. After the first dose 48 percent had nAbs, which increased to 97 percent after the second dose; and GMTs for binding and nAbs both increased significantly after the second dose [133].

## 7. Subunit Vaccines

Protein-based subunit vaccines are another safe vaccine against SARS-CoV-2. To safely produce an immune response, these vaccines use innocuous fragments of proteins that resemble the COVID-19 S-proteins. Similar to other COVID-19 vaccine mechanisms, subunit vaccines induce immunological responses to the SARS-CoV-2 spike (S) protein.

The S protein, prior to binding receptors on the cell membrane, is in a metastable pre-fusion conformation that undergoes significant rearrangement during virus–cell fusion. The immune responses generated by vaccine often target the pre-fusion S protein as this is more protective and limits transmission. The S-protein is comprised of an amino-terminal S1 receptor-binding domain (RBD) and a carboxy-terminal S2 subunit. The S1 subunit is a popular target during vaccine development as this is the main region implicated in allowing viral entry. A polyclonal antibody directed against various other epitopes of the S protein other than the RBD may be effective at hindering viral attachment, giving extra neutralizing activity, and/or preventing post-attachment fusion, for example. A vaccination that targets many epitopes might also reduce the risk of immunological escape through mutation. To boost the stability of the vaccines, proteins, which are frequently unstable, are packaged within nanoparticles and adsorbed onto adjuvants [134].

Genetically engineered subunit vaccines, otherwise known as recombinant protein vaccines, are constructed by incorporating pathogenic microorganism target genes with a vector which is then injected into an industrial organism in order to express the antigen of interest. These proteins are extracted from the organisms and used in vaccines to elicit an immunological response. These vaccines’ antigenicity is intimately linked to their expression mechanisms. Currently, yeasts, insect cells, mammalian cells, and bacteria are the most common methods used to express the antigen to generate subunit vaccines. Antigen-presenting cells with strong inherent adjuvant activity take up this form of the vaccine, such as the SARS-CoV nucleocapsid protein subunit vaccine. They effectively generate a T and B cell-mediated adaptive immune response. Recombinant protein vaccines provide a high level of inherent safety and stability. Furthermore, they can be manufactured in large quantities, making them ideal for mass immunization programs. However, recombinant protein vaccines have a number of drawbacks, including poor immunogenicity, a short immunization time, a dependency on immunization timing, and adjuvant type. Currently, COVID-19 recombinant protein vaccines use the SARS-CoV-2 surface S protein as the target antigen. The nucleocapsid protein, on the other hand, is immunogenic and has been employed in the production of COVID-19 recombinant protein vaccines. Although there is a risk of antibody-dependent enhancement (ADE) with recombinant protein, which occurs when antibodies bind pathogens but cannot prevent infection and thus the risk of worsening the severity of COVID-19 through ADE is a possible stumbling block for antibody-based vaccinations and therapies. Recombinant protein vaccines, in particular, have the ability to generate both mucosal and humoral immunity. Immunization efficacy can be improved by combining DNA vaccines and recombinant protein vaccines [135].

### 7.1. NVX-CoV2373

The Novavax created NVX-CoV2373, has already passed various rounds of clinical trials. The vaccine is comprised of full-length recombinant glycoprotein designed with protease-resistant mutations at the S1/S2 cleavage sites and two proline substitutions to stabilize the protein in a pre-fusion shape to withstand proteolytic degradation with protease resistance mutations and to bind to ACE2 receptors with a high affinity with saponin-based adjuvant (Matrix-M1). The requirements for adjuvants are necessary for the host cell to uptake more protein and produce a long-term immune. Antibody to S-binding was observed 21 days after the first dose, with a significant increase after the second dose. After the first dose, there was some nAb present, with a considerable rise by 7 days after the second dose. Based on IFN, IL-2, and TNF production in response to S protein stimulation, CD4+ T cell responses were detectable by 7 days following the second dosage, with a strong bias towards a TH1 cell phenotype; negligible TH2 cell responses [128]. Novavax was reported to have an 89.7% efficacy in a trial involving the B.1.1.7 strain in the UK. However, in a smaller and less conclusive South African experiment, where the great majority of infections were caused by the B. 1.351 type that is proliferating there, its efficacy plummeted to 60% [106].

### 7.2. EpiVacCorona

EpiVacCorona was developed by the Russian VECTOR Virology Center. It is a peptide-based COVID-19 vaccine comprised of three chemically produced peptides joined to a large carrier protein. Conserved S-protein epitopes were selected and conjugated to a chimeric recombinant carrier protein (Table 2). A Phase II clinical trial in Russia showed antibodies in 100% of participants after vaccination and third phase clinical trials have been registered, though no results are yet available [136].

The vaccine contains three chemically manufactured small segments of the viral spike protein—peptides—that reflect protein areas bearing B-cell epitopes that should be recognized by the human immune system. It is essential that the peptides carried by the vaccine are conjugated to a carrier protein. This gene is chimeric due to its creation from the genes of two different organisms, one encoding a viral nucleocapsid protein and the other encoding a bacterial maltose-binding protein (MBP). The artificial chimeric gene is tagged with a polyhistidine tag and expressed in E. coli. Following purification, the protein is conjugated with three peptides with each chimeric protein only attaching to one of the peptides. This produces three different conjugated molecules, each with the chimeric protein and 1 of the 3 peptides [136].

### 7.3. MVC COVID-19 Vaccine

The MVC COVID-19 vaccine, or the Medigen COVID-19 vaccine, is a protein subunit vaccine produced jointly with Medigen Vaccine Biologics Corporation in Taiwan, Dynavax Technologies in the United States, and the National Institutes of Health (NIH) in the United States. The recombinant S-2P spike protein is used to make this vaccine (Table 2). It is adjuvanted with Dynavax’s CpG 1018, which was formerly used in an FDA-approved adult hepatitis B vaccine [137].

### 7.4. Zifivax

Zifivax, also known as ZF2001or ZF-UZ-VAC-2001, is a COVID-19 adjuvanted protein subunit vaccine developed by Chinese company Anhui Zhifei Longcom Biopharmaceutical (Table 2). In China, Ecuador, Malaysia, Pakistan, and Uzbekistan, the vaccine candidate is in Phase III studies with 29,000 people. ZF2001 is built on the same technology as other protein-based vaccines. It is given in three doses over the course of two months. The Longcom company announced that early data from a Chinese phase III study with 28,500 people in August 2021 showed an overall efficacy of 82 percent against disease of any severity. The Alpha variation had a 93% efficacy rate, while the Delta variant had a 78% efficacy rate [138].

### 7.5. Corbevax

Corbevax is a protein subunit COVID-19 vaccine developed in a collaboration between Indian pharmaceutical company Biological and Baylor College of Medicine in Houston, Texas, and Dynavax. For development and production, it has been licensed to Biological E. Limited (BioE), an Indian biopharmaceutical company. The vaccine contains a variant of the SARS-CoV-2 spike protein’s receptor-binding domain (RBD), as well as the adjuvants aluminum hydroxide gel and CpG 1018 (Table 2).

### 7.6. Soberana 02

Soberana 02, PastoCovac, or FINLAY-FR-2 is a COVID-19 vaccine developed by the Finlay Institute in Cuba and the Pasteur Institute of Iran. It is a conjugate vaccination that comes in two doses, with the doses given 28 days apart. There is a third dose booster option 56 days after the first dose. In Iran, it was approved for emergency use in June 2021, while in Cuba, it was allowed for children over the age of two in August 2021. The vaccination is given in two doses, with the second shot given four weeks following the first. After only two doses, Phase III clinical trials with 44,000 participants showed the vaccine had a 92.45% efficacy. The conjugate vaccination FINLAY-FR-2 was developed by FINLAY (Table 2). It is comprised of the SARS-CoV-2 spike protein’s receptor-binding domain chemically coupled to tetanus toxoid [139].

## 8. COVID-19 Vaccine Adjuvants

Immunologic adjuvants are substances used in combination with specific vaccine antigens that serve to enhance and prolong antigen-specific immune responses [140]. In viral infection, T-cell recruitment is the hallmark and favored immunological response. In the case of subunit vaccines for SARS-CoV-2, the T-cell response is the more significant marker of vaccine success, and adjuvants aid in eliciting a predominantly Th1-skewed immune response.

The five most common COVID-19 subunit vaccine adjuvants are alum, beta defensin, MF59, matrix-M, and CpG. B-defensin adjuvanted multi-epitope subunit vaccines with 28 epitopes (three from replicase, three from NSp1, two from envelope, five from membrane, six from nucleocapsid, and nine from spike proteins) have been developed. The molecular docking revealed strong binding affinities for TLR3 and TLR8. These vaccines could protect against a wide range of diseases, particularly emerging variations of concern (VOC) [141].

A study with the full S-Matrix-M adjuvanted vaccination (NVXCoV2373) found a higher titer of S-protein antibodies and CD4+ and CD8+ T cells, follicular CD4+ Th, and germinal center B cells in mice spleen [142]. The safety, effectiveness, and tolerability of the S-AS03, S-CpG/alum, and placebo groups received 3, 9, and 30 g doses at 21-day intervals in a phase 1 subunit COVID-19 vaccine (SCB-2019). When opposed to AS03, CpG is a comparatively safe compound. S-AS03 and S-CpG/alum both cause the development of neutralizing antibodies (NA). S-AS03, on the other hand, developed a faster neutralizing antibody than the S-CpG/alum group, demonstrating the adjuvants’ differences. In the two adjuvanted groups, helper T-cell immune responses were generated, but this was not the case in S-protein specific to the non-adjuvanted S-trimer COVID-19 vaccination. The preferred options were 9 g S-trimer-AS03 and 30 g S-trimer-CpG/alum, according to this dose discovery study [143]. Yang et al. conducted a phase 1 and 2 subunit vaccination trial (2021). Adverse effects were mild to moderate in both phases 1 and 2. The seroconversion rates of NA in phase 2 were 76 percent and 72 percent in the 25 g and 50 g dosage groups, respectively, 14 days after the second treatment. After 14 days, seroconversion rates in the 25 g and 50 g groups were 97 percent and 93 percent, respectively, in the third dosage schedule. Three consecutive 25 g dosage shots were shown to be safe and efficacious when given at 14-day intervals [138]. Alum is used as an adjuvant in this vaccine.

### 8.1. B Defensing

Several SARS-CoV-2 subunit vaccine experiments included B-defensin as a TLR3 agonist [144,145,146]. Defensins are cationic peptides that serve as antimicrobials and signaling molecules in human innate and epithelial cells [147,148]. β-defensin is the most abundant in most cells [147]. In human epithelial cells, three -defensins have been identified: human -defensin-1 (hBD1), hBD2, and hBD3 [148]. hBD3 has a significant effect on dendritic cell and T cell activation, polarization, and migration [149]. It stimulates the production of IFN- γ and the generation of innate and adaptive immune responses [150]. A study looked at the adjuvant role of hBD2 and found that it boosted antiviral molecule expression levels [151].

### 8.2. Alum, Emulsion and Liposome

Cationic adjuvant formulations (CAF01), alum, squalene oil in water emulsion system (SE), and spike protein antigen were tested in mice. CAF01 generated more B, Th, and CD4+ T cells than alum, according to the findings. CAF01 are liposome adjuvant with a delivery vehicle of dimethyl dioctadecyl ammonium bromide (DDA) and an immunomodulator of synthetic mycobacterial cord factor. CAF01 generated larger titers of IFN- and IL-17 in similar research, although alum adjuvanted vaccinations tended to have more IL-5, IL-10, and IL-13. Spike proteins prior to fusion (S-2P), S1, and RBD-based subunit vaccines induced nucleoside analogs (NA) independent of adjuvants, according to studies [152,153].

### 8.3. CpG Adjuvant

Unmethylated CG motifs are found in cytosine phosphate guanidine oligodeoxynucleotides (CpG ODNs), a prominent new adjuvant. B lymphocytes and plasmacytoid dendritic cells are activated by TLR9 [154]. This leads to the production of more antigens which boosts helper T0 cell response and encourages proinflammatory cytokines production. When the vaccination antigen and ODN are near together, the adjuvant effects of CpG ODNs improve. Three types of synthetic CpG ODNs have been identified structurally [155], notably the ‘K/B,’ ‘D/A,’ and ‘C’ type ODNs. Different immunoglobulin types are activated by each class. CpG ODN, as a whole, is a unique and suggested adjuvant that works by increasing TNF- and IL-6 production. CpG has also been shown to improve the surveillance capabilities of antigen presenting cells. CpG ODNs’ value are boosted even more by their ability to boost mucosal and systemic immunity [155,156,157]. The CpG pattern is uncommon in the SARS-CoV-2 genome, and trends have been observed where new strains are favoring fewer CpG genomes. The increased proportion of asymptomatic and mild cases could be linked to the decreased CpG motif. As a result, employing CpG ODN as an adjuvant could be a suitable way to improve immunogenicity while reducing toxicity [158].

### 8.4. SARS-CoV-2 Variants and Their Effect on Vaccines

SARS-CoV-2 is an RNA virus and the mutation rate is high compared to DNA virus.

It was noted that the rate of evolution of SARS-CoV-2 is about 1.1 X 10-3 substitutions/year/site [159].

Due to continuous mutations, several variants have been generated. Scientists noted mutations in the SARS-CoV-2 variants. Di Caro et al. (2021) [160] illustrated that above 12,000 mutations have been recorded in the sequences of this virus. However, several variants have been generated from time to time. Among the variants, significant variants have been marked as VOCs/VOIs by WHO, eCDC, or CDC. Among the variants, emerging variants are Alpha (B.1.1.7; originated from the UK), Gamma (P.1; originated from Brazil); Beta (B.1.351; originated from South Africa); Delta (B.1.617.2; originated from India); Omicron (B.1.1.529: Reported in multiple countries) [161,162,163]. Significant mutations in the variants have resulted in immune escape, antibody escape, and partly vaccine escape [164,165]. Moore and Offit have described the rising concern of SARS-CoV-2 variants against the developed vaccine [166].

Several significant mutations have been observed in P681R, K417N, N501Y, E484K, and D614G in the spike which significantly affects the efficiency of the vaccine and is responsible for the evasion. However, the first mutation was reported as D614G which was reported for the increase of infectivity, alteration of the spike receptor binding activity, and thus immune escape [167]. The mutation (D614G) was found in all VOIs and VOCs. Researchers have illustrated that the mutation has occurred through positive selection [168]. At the same time, due to the significant mutations of the variants, decreased efficiency in vaccine-elicited sera has been noted by the variants from time to time. A recent study has created a pseudovirus with different mutations in the spike of Alpha variant (B.1.1.7) which resulted in reduced efficiency of vaccine sera. In this case, the inserted mutations in the pseudovirus are D1118H, del69/70, T716I, P681H, S982A, A570D, del 144/145, and N501Y [169]. In this study, mRNA-based vaccine BNT162b2 was used to assess the immune responses.

At the same time, a study assessed the B.1.1.7 and B.1.351 variants’ efficiency of neutralization by convalescent plasma. It has been noted that B.1.351 variant is resistant to sera from persons who have been vaccinated (about 10.3- to 12.4-fold). At the same time, B.1.351 variant showed a 9.4-fold reduction in neutralization by convalescent plasma [170].

Another recent study noted T-cell responses and neutralizing antibodies can be targeted by VOCs such as 1.1.529 (Omicron), B.1.617.2 (Delta), B.1.351 (Beta), and D614G (wildtype, WT). In a study, 60 healthcare workers were immunized using BNT162b2, or mRNA-1273, Ad26.COV2.S, ChAdOx-1 S. The study noted that neutralization assays with Beta and Delta showed consistent cross-neutralization. On the other hand, neutralization activity by the Omicron variant was absent or significantly less [171].

## 9. Conclusions

The global outbreak of SARS-CoV-2 necessitated the mobilization of researchers from many nations to collaborate and discover a solution to alleviate the effects of the pandemic. Early in these efforts, vaccines were identified as a promising solution and the COVID-19 vaccination efforts over the past two years have been significant for both the reduction of viral spread and the development of future vaccines for existing and emerging pathogens due to the considerable and rapid research efforts. This is possible owing to the availability of information from past pandemics of similar viruses in the cases of the SARS and MERS pandemics. Though many vaccine candidates were considered, today mainly mRNA and adenoviral vector vaccines have been administered to curb the effects of the pandemic. As of 24 May 2022, 4.71 billion individuals have been fully vaccinated, though only 15.9% of these individuals reside in low-income countries [172,173]. Although vaccines with high efficacy have been identified and are used today, further work into assuring that their storage needs can be met in low-income areas is essential to eradicating COVID-19.

Currently, new strains of SARS-CoV-2 continue to be identified, including the more recent subvariant of the Omicron variant, BA.2.12.1, and the continued monitoring of the virulence of these strains and the efficacies of currently approved vaccines against these emerging mutants is ongoing [174]. The possibility of introducing yearly boosters to currently vaccinated individuals is one proposed solution to combat the growing prevalence of these new strains [175].

## Figures and Tables

**Figure 1 biomedicines-10-01464-f001:**
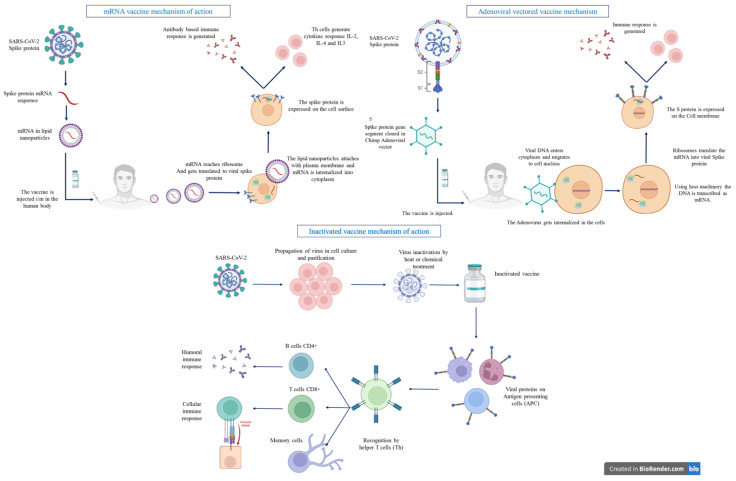
The mechanism of action by which immune response is generated by mRNA vaccine, adenoviral vaccines, and inactivated vaccines. Figure created using Biorender (https://biorender.com/ accessed on 6 April 2022).

**Table 2 biomedicines-10-01464-t002:** SARS-CoV-2 vaccine candidate information.

Sr. No.	Vaccine	Composition	Antigen Used	Dose	Storage Conditions
mRNA vaccine					
1	Pfizer BioNTech	Lipid nanoparticle (LNP) coupled mRNA 0.43 mg ALC-0315 = (4-hydroxybutyl) azanediyl) bis (hexane-6, 1-diyl) bis(2-hexyldecanoate), 0.05 mg ALC-0159 = 2-[(polyethylene glycol)-2000]-N,N ditetradecylacetamide, 0.09 mg 1,2-Distearoyl-*sn*-glycero-3-phosphocholine (DSPC), 0.2 mg Cholesterol, 0.01 mg Potassium dihydrogen phosphate, 0.07 mg Disodium hydrogen phosphate dihydrate pH 7–8 [11].	Spike (S) protein gene full length	Two shots with a gap of 21 days	−70 °C (long term) 2–8 °C up to 5 days
2	Moderna	LNP encapsulated mRNA. The core of LNPs contains mRNA, an ionizable cationic lipid, a neutral lipid and water. SM-102 (heptadecan-9-yl 8-((2-hydroxyethyl) (6-oxo-6-(undecyloxy) hexyl) amino) octanoate}, PEG2000-DMG = 1-monomethoxypolyethyleneglycol-2,3-dimyristylglycerol with polyethylene glycol of average molecular weight 2000, 1,2-Distearoyl-*sn*-glycero-3 phosphocholine (DSPC), Cholesterol, Tris (tromethamine) pH 7–8 [12].	Spike (S) protein gene full length	Two shots with a gap of 28 days	−20 °C (long term) 2–8 °C up to 30 days
Viral vectored vaccine					
3	Astra—Zeneca Oxford	Adenovirus: Viral vector (Replication deficient) encoding the SARS-CoV-2 Spike glycoprotein (ChAdOx1-S) not less than 2.5 × 10^8^ infectious units (Inf.U). Origin: Chimpanzee Adenovirus, L-Histidine, L-Histidine hydrochloride monohydrate, Magnesium chloride hexahydrate, Polysorbate 80 (E 433), Ethanol, Sucrose, Sodium chloride, Disodium edetate (dihydrate), Water for injections [13].	Spike (S) glycoprotein in the trimeric pre-fusion conformation	Two shots with a gap of 12 weeks	2–7 °C up to 6 months
4	Janssen Johnson & Johnson	Adenovirus: Viral vector (Replication incompetent) expressing the SARS-CoV-2 spike protein, Origin: Human Adenovirus serotype 26, Citric acid monohydrate, trisodium citrate dihydrate, ethanol, 2-hydroxypropyl-β-cyclodextrin (HBCD), polysorbate-80, sodium chloride [14].	Spike (S) glycoprotein	Only one shot	2–8 °C up to 3 months
5	Sputnik V	Two Recombinant Adenovirus - Human Adenovirus serotype 26 - Human Adenovirus serotype 5 Containing the SARS-CoV-2 S protein gene, in the amount of (1.0 ± 0.5) × 10^11^ particles per 0.5 mL dose. Tris-(hydroxymethyl)-aminomethane, Sodium chloride, Sucrose, Magnesium chloride hexahydrate, Disodium EDTA dihydrate, Polysorbate 80, Ethanol, and Water [15].	Unmodified full-length S-protein	Two shots with a gap of 21 days	−18 °C (liquid form) −2–8 °C (dry form)
Inactivated vaccines					
6	CoronaVac	Inactivated SARS-CoV-2 Virus (CZ02 strain) propagated on Vero cell line. 600SU per 0.5 mL. Adjuvant: Aluminum hydroxide Disodium hydrogen phosphate dodecahydrate, Sodium dihydrogen phosphate monohydrate, Sodium chloride [16].	Beta-propiolactone inactivated SARS-CoV-2 whole virus	Two shots with a gap of 14 days	2–8 °C
7	Covaxin	Inactivated SARS-CoV-2 Virus (NIV-2020-770 strain) 6 µg of whole-virion inactivated SARS-CoV-2 Adjuvant: Aluminum hydroxide gel (250 µg) TLR 7/8 agonist (imidazoquinolinone) 15 µg, 2-phenoxyethanol 2.5 mg, Phosphate buffer saline up to 0.5 mL [17].	Whole-virion inactivated SARS-CoV-2 antigen (Beta-propiolactone)	Two shots with a gap of 4 weeks	2–8 °C
8	Sinopharm BBIBP-CorV	Inactivated SARS-CoV-2 Virus (WIV04 strain) propagated on Vero cell line. 600SU per 0.5 mL. Adjuvant: Aluminum hydroxide Disodium hydrogen phosphate, Sodium dihydrogen phosphate monohydrate, Sodium chloride [18].	Beta-propiolactone inactivated SARS-CoV-2 whole virus (1:4000 vol/vol at 2 to 8 °C) for 48 h	Two shots with a gap of 21–28 days	2–8 °C up to 6 months (No freezing)
Subunit vaccine					
9	Epivac	Chemically synthesized peptide antigens of SARS-CoV-2 Spike protein. CRLFRKSNLKPFERDISTEIYQAGS, CKEIDRLNEVAKNLNESLIDLQE, CKNLNESLIDLQELGKYEQYIK Carrier protein: Chimera of Viral nucleocapsid protein and Maltose binding protein Adjuvant: Aluminium Hydroxide [20].	Chemically synthesized peptide antigens of SARS-CoV-2 Spike protein	Two shots with a gap of 21–28 days	2–8 °C
10	MVC COVID-19 vaccine	CHO cell derived spike protein (Subunit) Liquid, S-2P protein (15 mcg) + CpG 1018 (750 mcg) + Al(OH)_3_ (375 mcg) in 0.5 mL [21].	S-2P protein	Two shots with a gap of 4 weeks	2–8 °C
11	Zifivax	ZifiVax ZF2001 (ZF-UZ-VAC-2001) is a protein subunit vaccine using a dimeric form of the receptor-binding domain (RBD) as the antigen. 25 μg or 50 μg per 0·5 mL in a vial. Adjuvant: Aluminum hydroxide [22].	SARS-CoV-2 RBD antigen (residues 319–537, accession no. YP_009724390) Two copies in tandem repeat dimeric form. Manufactured in the CHOZN CHO K1 cell line	Three shots of 25 μg with a gap of 30 days	2–8 °C
12	CorbeVax	Spike receptor-binding domain (RBD) Adjuvant: Alhydroge (Alum) and CpG 1018 [24].	SARS-CoV-2 Spike protein RBD N1C1	Two shots with a gap of 28 days	2–8 °C
13	Soberana 02	Conjugate vaccine in which the virus antigen, the receptor-binding domain (RBD), is chemically bound to the tetanus toxoid [25].	SARS-CoV-2 Spike protein RBD	Two shots with a gap of 28 days	2–8 °C

## Data Availability

Not applicable.

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
