# Peer review of "Immune Response to SARS-CoV-2 Vaccines"

_biomedicines, 2022, doi:10.3390/biomedicines10071464_

Round 1

Reviewer 1 Report

This article well describes the immune response to the SARS-CoV-2 vaccine. However, there may be minor corrections and supplements necessary for the manuscrpt. Detailed comments and suggestion are listed below.

Minor

  • The authors described current vaccines, vaccine candidates and adjuvants. It would be better for the reader`s understanding, if they could provide a table for this contents.
  • It is necessary to supplement the conclusion of the thesis.
  • Check the spacing of the first line of the sentence.
  • It would be good to categorize the content numerically. Ex) 1. Inactivated viral vaccine candidates~, 1) Corona Vac; 2) BBIBP-CorV etc...

Reviewer 2 Report

The review is an oversight on SARS-CoV-2 vaccines. This kind of overview is interesting to clear such an important subject in continuous evolution. The review is overall objective, well written and relatively easy to read. I have some major suggestions to make before publication:
-Chapter organization is insufficient. After the introduction there is non number coding. 

-It is necessary to introduce at least 1 figure and 1 tabella especially in the section of vaccine description efficacy and use

-The review is missing a conclusion and future highlights on vaccine use and management 

Round 2

Reviewer 2 Report

The authors have faced the highlighted issues according to suggestions. The article is now suitable for publication.